# Theoretical and Experimental Study on the Thermal Insulation Performance of the Roof with Water-Retained Bricks

**DOI:** 10.3390/e24111528

**Published:** 2022-10-25

**Authors:** Rubing Han, Zhimao Xu, Enshen Long

**Affiliations:** 1College of Architecture and Environment, Sichuan University, Chengdu 610065, China; 2School of Civil Engineering and Architecture, Southwest University of Science and Technology, Mianyang 621010, China; 3Sichuan Electric Power Design Consulting Co., Ltd., Chengdu 610016, China

**Keywords:** theoretical and experimental study, thermal insulation performance, roof, water-retained brick

## Abstract

In this paper, the thermal insulation performance of the roof with water-retained bricks was first analyzed theoretically with respect to the thermal inertia, attenuation and delay time of the roof with water-retained bricks. Then, the experimental rig was established to carry out the experimental research on the thermal insulation performance of the roof with and without water-retained bricks on the sunny, overcast and rainy days in the summer and on the sunny day in the winter. The results showed that: (1) the surface heat storage coefficient is affected by the evaporating heat transfer of the water layer; (2) the thermal inertness, attenuation and delay time of the roof with water-retained bricks are 2.575, 21 and 6.94 h, respectively, when the water depth is 2 cm; (3) on the sunny, overcast and rainy days in the summer, laying water-retained bricks can enhance the heat insulation performance of the roof, and can improve the thermal comfort of the loft; and (4) on the sunny day in the winter, after laying water-retained bricks, the average temperature of the loft in 24 h increases by 2.3 °C, and the temperature fluctuation of the loft decreases by 56.0%. Therefore, the thermal insulation effect is significantly improved after laying water-retained bricks on the roof from the results of both the theoretical and experimental study.

## 1. Introduction

The percentage of urbanization will reach about 66% by 2050 [1], which will lead to serious issues such as urban heat island (UHI) effect [2]. UHI effect contributes more challenged living conditions for human beings [3,4]. In this circumstance, more and more energy-efficient building measures such as green roofs and roof ponds are born at the right moment [5,6] and thus the UHI effect can be partially alleviated [7].

Recently, green roofs have been investigated by some scholars since they have lightweight and good thermal insulation performance. Chen et al. [8] established a radiative roof cooling performance model to conduct thorough parametric studies on pivotal factors governing achievable equilibrium temperature and net cooling power in building roofs. The results showed that compared with broadband- and transmittance-based adaptive emitters as roof surfaces, an atmospheric-window-based selective emitter can achieve the minimum sub-ambient equilibrium temperature. Mishra et al. [9] carried out a green roof design and cost estimation for the main building, new boys’ hostel and new building block at the Amity University campus. The results showed that the total runoff reduction is the maximum in the new building roof area. The water retained due to green roof will help in reducing the temperature and also improve the air quality in the surrounding area. Schreiber et al. [10] systematically analyzed green roof issues of Canadian buildings. Air tightness, attic ventilation, and insulation requirement are the three factors that mainly influence the performance of green roofs. Baskar et al. [11] proposed a real time building model to understand the efficiency in reducing the internal temperature. The results showed that the developed 2-D finite element model was able to accurately predict the exterior and interior surface temperatures of both models. Yuan et al. [12] investigated the thermal characteristics of green roofs installed on residential buildings in Qatar’s hot and dry climate in order to assess their viability and determine how best to minimize energy usage. The results indicated that the plant leaf area index is the plant planting index that works best in this environment, and the height of the plant substrate layer is 23 cm, and the height of the cultivation layer is 18 cm. Susca et al. [13] provided a geographically explicit review of the potential building energy benefits deriving from the installation of green roofs depending on their specific design. The results showed that green roof deployment is beneficial in mitigating UHI at the rooftop level in all the climate areas investigated, and the thermal insulation provided by the soil layer of green roofs is crucial in decreasing building energy demand in cold climates. Feng et al. [14] studied the optimized combination of the shortwave absorptivity and longwave emissivity for exterior roof surfaces. The results showed that for cold areas and seasons, a combination of high shortwave absorptivity and low longwave emissivity is preferred. Abdalazeem et al. [15] investigated the effect of GR (157 papers) and PV/GR systems (28 papers) that addressed UHI mitigation and how energy-saving and indoor thermal comfort was achieved in urban buildings. It was found that GR and GR/PV systems have a positive impact on improving dominant parameters in hot arid climates, especially on the building scale. Bevilacqua et al. [16] investigated the thermal performances of different green roof solutions in the attempt to assess the effect on the building energy demand. Some aspects (such as UHI mitigation and hydrological management) can become even more relevant in an optic of an adequate and sustainable planning of urban areas. Cascone et al. [17] designed an innovative and sustainable green roof and it was compared to two traditional green roofs and to the existing roof. The results demonstrated that the green roof reached higher surface temperatures and maintained an almost constant volumetric water content. Xue et al. [18] studied the composition and the preparation process of a bi-functional cool white roof coating and the composition and the preparation process of a bi-functional cool white roof coating. The results showed that the artificial accelerated weather resistance of the cool white roof coating based on styrene acrylate copolymer and cement is qualitatively good.

In addition, many researchers carried out the thermal performance of the buildings with using the roof pools. Scholz et al. [19] evaluated the water treatment potential of a stormwater pond system after 15 months of operation. The system was based on a combined silt trap, attenuation pond and vegetated infiltration basin. The treatment of rainwater runoff from roofs was found to be largely unnecessary for recycling. Blaauwendraad et al. [20] proposed a method for analyzing rainwater accumulation on flat or nearly flat one-way and two-way roofs. The water storage capacity of deformed secondary components and steel plate units was estimated. Spanaki et al. [21] investigated covered/uncovered pond with/without sprays, focusing on the comparison of characteristic roof pond variants, and the advantages and disadvantages of ponds are discussed with respect to appropriate decisions on roof pond variants for cooling purposes. Tang et al. [22] developed a detailed simulation model for the study and analysis of rooftop pools with surface oxidation of jute bags. The simulation results showed that it has better cooling performance than the wet gunny-covered roof in terms of both indoor air temperature and heat entering the pond through the roof. Blaauwendraad et al. [23] proposed a piston-spring model for rigid roofs and a rod-spring model for flexible roofs to check the sensitivity of roofs to water. The derived theory clarified the effect of the profiled steel roof sheeting on safety and other practical hints were included. Carlos et al. [24] investigated a novel wet fabric device for indirect evaporative cooling of the roof, with a theoretical model describing the temperature of the interior, roof, water, and fabric, considering constant properties. The results showed that the proposed wet fabric device has cooling potential under three climatic conditions, and the fabric porosity has a significant effect on the internal temperature. Kharrufa et al. [25] used a one-space building to test the effects of a rooftop pond that is mechanically ventilated for summer cooling. Results showed that the indoor temperature is significantly improved, and the peak time of outdoor temperature is significantly reduced at 3 o’ clock. Sharifi et al. [26] systematically reviewed the literature on the use of rooftop pools for passive heating and cooling of buildings. This review showed that rooftop pools can provide year-round thermal comfort while reducing the need for active heating and cooling systems. Almodovar et al. [27] studied two roof pool structures combined with a water-air heat exchanger. Results demonstrated that the cells with roof ponds have better cooling performance than the code compliant control cell.

However, no literature mentioned the theoretical and experimental study on the thermal insulation performance of the roof with water-retained bricks as illustrated in Figure 1. Figure 1 is a cross-sectional schematic of the roof with water-retained bricks. The brick is equipped with a movable cover with a strip-hole. The thickness of the cover is 2 cm. Considering the strength of the cover, the distance between two strip-holes is set to 6 cm, and the length of the strip-holes is set at 18 cm, and both ends of the strip-holes are semicircles. The upper layer of the brick stores water, and the lower part is combined with the roof to form an air layer. Moreover, considering the building modulus, the transportation of water-retained bricks and the convenience of construction, the horizontal size of the water-retained bricks is set to 30 × 30 cm. A certain distance is maintained between the side wall of the cofferdam and the cover, which is conducive to ventilation and evaporation. As shown in the figure, above the roof, there are some layers such as the air interlayer, lid, water layer, air layer and cover.

The following investigations will be performed in this study:Theoretical calculation of the thermal insulation performance of roof with water-retained bricks;Establishment of the experimental rig of testing the thermal insulation performance of roof with water-retained bricks;Experimental investigation on the thermal insulation performance of the roof with and without water-retained bricks on the sunny, overcast and rainy days in the summer;Experimental investigation on the thermal insulation performance of the roof with and without water-retained bricks on the sunny in the summer.

## 2. Theoretical Analysis of the Thermal Insulation Performance

### 2.1. Heat Storage Coefficient of Each Layer of the Roof with Water-Retained Bricks

The following equation is used to express the heat storage coefficient of each layer:(1)$$S_{24}=0.27\sqrt{\lambda \cdot c\cdot \rho_{0}}$$
where: *S*_24_ is the thermal storage coefficient of each layer with a thermal action period of 24 h, W/(m^2^·K); *λ* is the thermal conductivity of each layer, W/(m·K); *c* is the specific heat capacity of each layer, kJ/(kg·°C); *ρ*_0_ is the density of each layer, kg/m^3^.

The measured thermal conductivity of the lid with ceramsite concrete is 0.502 W/(m·°C), its specific heat capacity is 1.05 kJ/(kg·°C), and its density is 1740 kg/m^3^. With Equation (1), the heat storage coefficient (*S*_24,lid_) of the lid with ceramsite concrete is 7.938 W/(m^2^·°C).·

*λ_air_* is the thermal conductivity of air, *λ_air_* = 2.63 × 10^−2^W/(m·°C), then *S*_24,air_ = 0.048 W/(m·°C), so the heat storage coefficient of air is thought as 0.

The cover is made of heterogeneous material, and the thermal resistance caused by strip holes is transferred to the water surface interface. Therefore, the average thermal conductivity of the cover is calculated with the following equation:(2)$${\bar{\lambda }}_{lid}=\frac{\lambda_{lid}\cdot A_{lid}+\lambda_{air}A_{opening}}{A_{lid}+A_{opening}}$$
where: *λ_lid_* is the thermal conductivity of ceramsite concrete, w/(m·°C); *A_lid_* is the area of the perforated cover, m^2^; *A_opening_* is the area of strip holes on the cover, m^2^. The cover of the water-retained brick under the study exists $\frac{A_{lid}}{A}\approx 0.75$, thus, ${\bar{\lambda }}_{lid}=0.75\lambda_{lid}+0.25\lambda_{air}$, calculated ${\bar{\lambda }}_{lid}$ = 0.0376 W/(m·°C); the thickness of the cover *d* = 0.02 m, ${\bar{R}}_{lid}=\frac{d}{{\bar{\lambda }}_{lid}}$ = 0.053 (m^2^·°C)/W, ignoring the air heat storage coefficient, then, ${\bar{S}}_{lid}\approx 0.75S_{lid}$ = 5.953 W/(m^2^·°C).

The average thermal resistance of the water layer is ${\bar{R}}_{w}=\frac{H}{{\bar{\lambda }}_{w}}$, where *H* is the depth of water and ${\bar{\lambda }}_{w}$ is the average thermal conductivity of the water, which is 60 × 10^−2^W/(m·°C). The specific heat of the water at 20 °C is 4.18 kJ/(kg·°C), and the density of the water at 20 °C is 998 kg/m^3^, thus the average thermal storage coefficient of water layer is 13.5 W/(m^2^·°C).

The upper part of the water layer is saturated water vapor near the water surface, and the part far away from the water surface is humid air. The ${\bar{\lambda }}_{vapor}$ of saturated water vapor is 0.25 W/(m·°C), and the *ρ_vapor_* of saturated water vapor at 20 °C is 0.017kg/m^3^. Then, ${\bar{S}}_{24,vapor}$ = 0.036 W/(m^2^·°C).

The fluctuation of surface temperature is not only related to the thermal and physical properties of each layer material, but also related to the boundary conditions. The surface heat transfer performance related to boundary conditions is expressed by surface heat storage coefficient of *Y*, which has the same physical meaning and definition as with *S*, but has a different calculation equation. When *D* ≥ 1.0, the effect of boundary conditions is negligible, *Y≈S*. When *D <* 1.0, the influence of boundary conditions on the other side of the layer cannot be ignored, then *Y≠S*.

Taking the calculation of external surface heat storage coefficient as an example, the temperature harmonic wave is transmitted from outside to inside. The heat storage coefficient of *Y*_1*,e*_ for the outer surface of the first layer can be divided into two cases: If *D*_1_ ≥ 1.0, *Y*_1*,e*_
*= S*_1_; If *D*_1_
*<* 1.0, $Y_{1,e}=\frac{R_{1}S_{1}^{2}+h_{i}}{1+R_{1}h_{i}}$. As for the other layers, it is necessary to judge the thermal inertia of the layer every time. If *D_m_* ≥ 1.0 in the layer of *m*, *Y*_m,e_ = S_m_, and if *D_m_ <* 1.0, $Y_{m,e}=\frac{R_{m}S_{m}^{2}+Y_{m-1,e}}{1+R_{m}Y_{m-1}{}_{,e}}$ (*m* = 2, 3, 4,…, *n*). The thermal storage coefficient of the outer surface of the outermost layer is the thermal storage coefficient of the outer surface of the flat wall, that is *Y_n,e_ = Y_ef._*. When calculating the heat storage coefficient of the inner surface, the calculation process is the same.

### 2.2. Thermal Inertia of the Roof with Water-Retained Bricks

The thermal inertia of the roof with water-retained bricks can be expressed as follows:(3)$$D=\sum_{i=1}^{n}D_{i}=\sum_{i=1}^{n}R_{i}S_{i}$$
where: *D* is the thermal inertia of the roof with water-retained bricks, which is dimensionless; *D_i_* is the thermal inertia of the layer *i*; *R_i_* is the thermal resistance of the layer *i*, (m^2^·°C)/W; *S_i_* is the heat storage coefficient of the layer *i*, W/(m^2^·°C).

### 2.3. Attenuation and Delay Time of Temperature Harmonics

The attenuation of temperature harmonics of the roof with water-retained bricks can be expressed by the following equation:(4)$$\upsilon_{0}=0.9e^{\frac{\Sigma D}{\sqrt{2}}}\cdot \frac{S_{1}+h_{i}}{S_{1}+Y_{1}}\cdot \frac{S_{2}+Y_{1}}{S_{2}+Y_{2}}\cdot \cdot \cdot \cdot \cdot \cdot \frac{S_{n}+Y_{n-1}}{S_{n}+Y_{n}}\cdot \frac{Y_{n}+h_{o}}{h_{o}}$$
where: *ν*_0_ is the total attenuation of the roof with water-retained bricks, which is dimensionless; *∑D* is the sum of thermal inertia; *S_1,_ S_2_…S_n_* is the heat storage coefficient of each layer, W/(m^2^·°C); *Y_1,_ Y_2_…Y_n_* is the heat storage coefficient of each layer, W/(m^2^·°C); *h_i_* is the heat transfer coefficient of the inner surface of the roof with water-retained bricks, W/(m^2^·°C); *h_o_* is the heat transfer coefficient of the outer surface of the cover, W/(m^2^·°C).

The total delay time of temperature harmonics of the roof with water-retained brick is *ξ*_0_, which can be expressed by the following equation:(5)$$\xi_{0}=\frac{1}{15}(40.5\sum D+\arctan\frac{Y_{ef}}{Y_{ef}+h_{o}\sqrt{2}}-\arctan\frac{h_{i}}{h_{i}+Y_{if}\sqrt{2}})$$
where: *Y*_ef_ is the heat storage coefficient of the outer surface, W/(m^2^·°C); *Y*_if_ is the heat storage coefficient of the inner surface, W/(m^2^·°C).

Table 1 is the thermal and physical parameters of each layer of the roof with water-retained bricks, in which, the layers are numbered from inside to outside. *h_o_* = 19.0 W/(m^2^·°C) and *R_o_* = 0.05 (m^2^·°C)/W of the outer surface; *h_i_* is 8.7 W/(m^2^·°C) and *R_i_* = 0.11 (m^2^·°C)/W of the inner surface. The outer surface heat storage coefficient *Y_e_* of each layer from inside to outside is calculated as follows:①*D*_1_ < 1.0, $Y_{1,e}=\frac{R_{1}S_{1}^{2}+h_{i}}{1+R_{1}h_{i}}=9.228$;②*D*_2_ < 1.0, $Y_{2,e}=\frac{R_{2}S_{2}^{2}+Y_{1,e}}{1+R_{2}Y_{1,e}}=11.050$;③*D*_3_ < 1.0, $Y_{3,e}=\frac{R_{3}S_{3}^{2}+Y_{2,e}}{1+R_{3}Y_{2,e}}=11.178$;④*R*_4_ = 0.14 (m^2^·°C)/W, *Y*_4,*e*_ = 4.358 W/(m^2^·°C);⑤*D*_5_ < 1.0, $Y_{5,e}=\frac{R_{5}S_{5}^{2}+Y_{4,e}}{1+R_{5}Y_{4,e}}=3.041$;⑥*D*_6_ > 1.0, *Y*_6,e_ = S_6_ = 13.5;⑦*D*_7_ < 1.0, $Y_{7,e}=\frac{R_{7}S_{7}^{2}+Y_{6,e}}{1+R_{7}Y_{6,e}}=5.92$;⑧*D*_8_ < 1.0, $Y_{8,e}=\frac{R_{8}S_{8}^{2}+Y_{7,e}}{1+R_{8}Y_{7,e}}=5.94$.

The total attenuation of the roof with water-retained bricks is:
(6)$$\upsilon_{0}=0.9e^{\frac{\Sigma D}{\sqrt{2}}}\cdot \frac{S_{1}+h_{i}}{S_{1}+Y_{1}}\cdot \frac{S_{2}+Y_{1}}{S_{2}+Y_{2}}\cdot \frac{S_{3}+Y_{2}}{S_{3}+Y_{3}}\cdot \frac{S_{4}+Y_{3}}{S_{4}+Y_{4}}\cdot \frac{S_{5}+Y_{4}}{S_{5}+Y_{5}}\cdot \frac{S_{6}+Y_{5}}{S_{6}+Y_{6}}\cdot \frac{S_{7}+Y_{6}}{S_{7}+Y_{7}}\cdot \frac{S_{8}+Y_{7}}{S_{8}+Y_{8}}\cdot \frac{Y_{8}+h_{o}}{h_{o}}$$

By using the above-mentioned formula, the calculated thermal inertia, attenuation and delay time of roof with water-retained brick under different water depths of 0, 2, 4, 6, 8, 10, 12 cm are presented in Table 2.

According to the data in Table 2, the relations between *D, ν*_0_ and *ξ*_0_ with *H* can be fitted as below:*D* = 0.224*H* + 2.127(7)
*ν*_0_ = 10.96e^0.334*H*^(8)
*ξ*_0_ = 1.211*H* + 4.524 (9)

When the water depth is 0 cm, the attenuation is the smallest with value of 10.96. The attenuation is most affected by the water depth, and the larger the water depth, the faster the attenuation grows. When the water depth is 0 cm, the delay time is the smallest with a value of 4.524. With the increase in water depth, the delay time increases. If the delay time is controlled within 8–10 h, the water depth should not exceed 8 cm.

Therefore, *D*, *ν*_0_ and *ξ*_0_ increase with the increase in H. The slope of ξ_0_ is 5–6 times that of D. For every 1 cm increase in h, the thermal inertia will increase by 0.224, while the delay time will increase by 1.2 h. Therefore, the delay is more affected by water depth than the thermal inertia. The increase in the delay time will transfer the high temperature of the day to the night. Since the outdoor air temperature at night is low, it can be cooled by the well-organized natural ventilation in the room. Therefore, the appropriate increase in the delay time is conducive to reducing the room temperature during the day. However, the water depth is too large, the delay time is too large, the heat accumulated in each layer of the roof during the day continues to transfer indoors, and if the ventilation is bad, it is bound to lead to stuffy at night, the opposite.

According to calculated values of the thermal inertia, attenuation and delay time, it can be indicated that the roof with proposed water-retained bricks has evident thermal insulation characteristics. Hence, the theoretical calculation shows that laying water-retained bricks can improve the thermal insulation performance of the roof. The following experiment will be carried out to verify its thermal insulation performance in different seasons and different weathers.

## 3. Experimental Scheme for Testing the Insulation Performance

The tested building is a five-story dormitory building with north–south orientation on a high ground in Mianyang city, Sichuan Province. There are no tall buildings or trees around to block the solar radiation; internal heat disturbance and external disturbance caused by irregular opening and closing of doors and windows are purposely eliminated during the test.

The main tested parameters are: meteorological parameters such as atmospheric pressure (*P*), solar radiation intensity (*I*), wind speed (*v*), air temperature (AT), relative humidity (R.H.) of the air; bottom temperature of the brick (BT), roof temperature under the brick (RTWB), roof temperature (RT), water temperature (WT), ceiling temperature of the rooftop without bricks (CT), and ceiling temperature of the rooftop with bricks (CTWB) which can be considered as one of indicators that influence indoor thermal comfort.

Figure 2 and Figure 3 show the schematic and photo of the field test, respectively. As shown in Figure 2, the water-retained brick is equipped with a movable cover with a strip-hole. The upper layer of the brick stores water, and the lower part is combined with the roof to form an air layer. Moreover, considering the building modulus, the transportation of water-retained bricks and the convenience of construction, the horizontal size of the water-retained bricks is set to 30 × 30 cm. The resolution, range and accuracy of the sensors utilized are given in Table 3.

## 4. Experimental Results and Discussion

### 4.1. In Summer

The summer experimental period is from August 1, 2015 to August 17, 2015, and August 6 is the day of continuous clear weather, which is used to analyze the influence of meteorological parameters on the thermal insulation performance of the roof with and without water-retained bricks under clear weather conditions. August 14 is overcast, which is used to analyze the influence of meteorological parameters on the thermal insulation performance of the roof with and without water-retained bricks under overcast conditions. August 16 to 17 are rainy days, both of them are utilized to identify the influence of meteorological parameters during rainy days on the thermal insulation performance of the roof with and without water-retained bricks.

(1)
**
*On a Sunny Day*
**


The meteorological parameters of the sunny day of August 6, 2015 are shown in Figure 4. The maximum solar radiation intensity (*I*) is 1074 W/m^2^, the average solar radiation intensity is 526 W/m^2^ from 7:00 to 19:00, and the average solar radiation intensity is 665 W/m^2^ during the time when the sunlight can completely incident on the water surface (9:00 to 15:00). The average air temperature (AT) of 24 h is 28.8 °C, the highest AT is 35.2 °C, and the lowest AT is 22.5 °C. The maximum deviation of AT within 24 h is 6.7 °C. The average R.H. in 24 h is 59.7%, the maximum R.H. is 81.7%, and the minimum R.H. is 35.3%. According to the results, it can be observed that the AT is apparently influenced by the solar radiation intensity to some extent.

Figure 5 shows the measured values of wind speed (*v*) and roof temperature under the brick (RTWB) on the same sunny day in the summer. As can be seen from the figure, the maximum wind speed is 1.6 m/s, and the instantaneous fluctuation of the wind speed is relatively large. There is no obvious correlation between the wind speed and RTWB during the test period.

Figure 6 shows the measured values of the solar radiation intensity (*I*) and the roof temperature under the brick (RTWB) on the same sunny day in the summer. As can be seen from the figure, RTWB is greatly influenced by *I*, and RTWB gradually increases with the increase in *I*, and the time when the maximum value of RTWB appears is 5.1 h later than that of *I*.

Figure 7 shows the measured values of characteristic temperatures on the sunny day in the summer. Characteristic temperatures include water temperature (WT), bottom temperature of the brick (BT), roof temperature under the brick (RTWB) and the air temperature (AT). It can be seen that AT starts to rise from 7:00, but the water temperature still drops slowly at this time, and begins to rise until 9:00. BT at night is slightly lower than WT, while BT increases with WT during the day, and the temperature difference between the two is small. The difference between BT and WT reaches its maximum at about 17:00. From 11:40 to 21:10, BT is greater than RTWB, and this temperature profile determines the mode and direction of heat transfer. RTWB is lower than BT from 21:10, while BT is higher than the AT at night, and the heat dissipation capacity of the roof under the brick is weakened. In general, the trend of WT, BT and RTWB is similar to that of AT, that is to say WT, BT and RTWB are impacted by AT. However, there is a time delay for the three temperatures as compared to the AT. Compared with the ordinary roof, the heat dissipation capacity of the roof under the water-retained brick reduces, and thus the heat loss reduces.

Figure 8 illustrates the measured values of the roof temperature under the brick (RTWB) and roof temperature without the brick (RT) on the sunny day in the summer. As can be seen from the figure, the average RTWB within 24 h is 26.1 °C, the maximum RTWB is 28.0 °C, the minimum RTWB is 24.2 °C, and the maximum fluctuation of RTWB is 3.8 °C. The average RT in 24 h is 36.8 °C, the maximum RT is 56.3 °C, the minimum RT is 23.7 °C, and the maximum fluctuation of RT is 32.6 °C. After laying water-retained bricks, roof temperature fluctuation decreases by 88.3%. The ordinary roof has high daytime temperature and low night temperature, which is greatly affected by solar radiation and cold air radiation. Because of the influence of the upper water-retained brick, solar radiation and cold air radiation cannot directly exchange heat with the roof with water-retained bricks. Therefore, the temperature fluctuation is small and the average temperature is low.

Figure 9 displays the measured values of the ceiling temperature of the rooftop with bricks (CTWB) and ceiling temperature of the rooftop without bricks (CT) on the sunny day in the summer. As can be seen from the figure, the average CTWB of 24 h is 28.0 °C, the maximum CTWB is 29.7 °C, the minimum CTWB is 26.9 °C, and the maximum deviation of CTWB is 2.8 °C. In 24 h, the average CT is 28.6 °C, the maximum CT is 32 °C, the minimum CT is 26.1 °C, and the maximum deviation of CT is 5.9 °C. After laying water-retained bricks, the maximum fluctuation of the ceiling temperature of the rooftop can be reduced by 2.3 °C, and the fluctuation of the ceiling temperature of the rooftop can be reduced by 52.5%. The heat insulation effect is significantly improved, so the indoor thermal comfort is improved.

(2)
**
*On an Overcast Day*
**


Figure 10 illustrates the meteorological parameters on the overcast day of 14 August 2015. It can be seen from the figure that the maximum solar radiation intensity is 391 W/m^2^, the average solar radiation intensity from 7:00 to 19:00 is 107 W/m^2^, the average AT of 24 h is 24.7 °C, the highest AT is 26.7 °C, and the lowest AT is 22.7 °C. The average R.H. of 24 h is 90.3%, the maximum R.H. is 98.5%, and the minimum R.H. is 84.9%.

Figure 11 indicates the measured values of wind speed (*v*) and roof temperature under brick (RTWB) on the overcast day in the summer. It can be revealed that the maximum wind speed occurring between 11:00 and 11:50 is 2.1 m/s. From 8:40 to 9:50, the average wind speed is 1.1 m/s. The RTWB continues to drop, which is related to the continuous drop in AT. From the meteorological condition, the wind may lead to the drop in AT, but the influence of wind speed on the RTWB is not regular.

Figure 12 exhibits the RTWB values with solar radiation intensity (*I*) and roof temperature under brick (RTWB) on the overcast day in the summer. As can be seen from the figure, the small solar radiation intensity is equivalent to no external heat input on the roof, so the RTWB continues to decrease.

Figure 13 shows the measured values of characteristic temperatures on the overcast day in the summer. It can be observed that RTWB > BT > WT, which means that the roof continues to dissipate heat outwards. The temperature fluctuations of WT and BT are similar to those of air temperature, which indicates that the characteristic temperatures of WT and BT are greatly affected by AT. The fluctuation of RTWB is smaller than that of WT and BT, which implies that the temperature fluctuation is influenced by the temperature attenuation and delay of the roof with water-retained brick. In addition, the AT decreases first from 0:00 to 14:00, and then it increases with time until to 17:00, after then it decreases with time again; WT and BT have a similar change pattern, however, the peak time varies to 19:00 and 20:00, respectively.

Figure 14 presents the hourly values of RTWB and RT on the overcast day in the summer. As can be seen from the figure, RTWB continues to decrease within 24 h, whereas RT begins to rise at 14:00 after experiencing a decline because the influence of *I* on RT is significantly greater than that on RTWB. From 20:00 to 24:00, RT is always lower than RTWB, that is, RT < RTWB on the overcast night. In addition, the fluctuation of RTWB within 24 h is less than RT.

Figure 15 displays the measured values of the CTWB and the CT on the overcast day in the summer. As can be seen from the figure, the average CTWB of 24 h is 27.8 °C, the maximum CTWB is 29.0 °C, the minimum CTWB is 27.0 °C, and the maximum fluctuation of CTWB is 2 °C. In 24 h, the average CT is 28.0 °C, the maximum CT is 31.6 °C, the minimum CT is 26.5 °C, and the maximum fluctuation of CT is 5.1 °C. Moreover, the change in the rule of CTWB and CT with time is quite similar with each other. After laying water-retained bricks, the temperature fluctuation of the ceiling temperature of the rooftop drops by 60.0%, and hence the indoor thermal comfort is improved.

(3)
**
*On Rainy Days*
**


Figure 16 demonstrates the meteorological parameters of rainy days from August 16 to 17, 2015. It can be seen that the maximum solar radiation intensity (*I*) is 120 W/m^2^ on August 16 and 207 W/m^2^ on August 17.

Figure 17 displays the wind speed (*v*) and RTWB on the rainy days in the summer. As can be seen from the figure, the two days are windy, and the 10 min maximum average wind speed is 2.9 m/s, which is the intermittent gale weather. RTWB continues to decline.

Figure 18 illustrates the measured characteristic temperatures on rainy days in the summer. As can be seen from the figure, the change in RTWB, BT and WT with time is very similar with each other; however, with RTWB > BT > WT, the temperature everywhere shows a similar rule to that on the overcast day. However, the difference between WT and BT on rainy days is smaller to that on overcast days. This is because the water-retained bricks do not cover the whole roof, and the rainwater can flow to the roof through the gaps between the bricks and evaporate when being heated, while the bottom of the bricks prevents the vapor from spreading upward. Therefore, BT and WT are affected by rain similarly, and the temperature difference between them is small.

Figure 19 shows the measured values of roof temperature under the brick (RTWB) and roof temperature without brick (RT) on rainy days in the summer. As shown in the figure, the average RTWB over 48 h is 23.3 °C, the maximum RTWB is 25.9 °C, the minimum RTWB is 21.6 °C, and the maximum fluctuation of RTWB is 4.3 °C. The average RT of 48 h is 23.8 °C, the maximum RT is 29.7 °C, the minimum RT is 20.8 °C, and the maximum fluctuation of RT is 8.9 °C. After laying water-retained bricks, the roof temperature fluctuation decreases by 51.7%.

Figure 20 indicates the measured ceiling temperature of the rooftop without bricks (CT) and the ceiling temperature of the rooftop with bricks (CTWB) on rainy days. As can be seen from the figure, the average CTWB over 48 h is 25.3 °C, the maximum CTWB is 28.0 °C, the minimum CTWB is 22.9 °C, and the maximum fluctuation of CTWB is 5.1 °C. Additionally, the average CT over 48 h is 24.5 °C, the maximum CT is 29.8 °C, the minimum CT is 21.4 °C, and the maximum fluctuation of CT is 7.4 °C. After laying water-retained bricks, the fluctuation in the ceiling temperature of the rooftop decreases by 31.1%, and the indoor thermal comfort is also improved.

In conclusion, the roof temperature under the brick is greatly affected by solar radiation and air temperature. Compared with the ordinary roof, the surface temperature and inner roof temperature fluctuate less than that of the ordinary roof, and the thermal comfort is improved.

### 4.2. In Winter

The winter experiment was conducted from February 1 to 10, 2016. February 7 and 8 were sunny days and the data of February 8 was selected to analyze the influence of meteorological parameters on characteristic temperatures of roof with and without water-retained bricks under sunny weather conditions in the winter.

Figure 21 shows the meteorological parameters on February 8. It can be seen from the figure that the maximum solar radiation intensity is 663 W/m^2^, which appears at 13:00. The period with strong solar radiation intensity is 14:00 to 15:00. The night air temperature (AT) is low and the relative humidity is high; the daytime AT rises gradually with the increase in solar radiation intensity, the relative humidity drops from 90% at 10:00 to the minimum of 31.2% at 15:00, and then gradually rises. The 24 h average AT is 6.5 °C, the maximum AT is 11.8 °C, appearing at 14:00; the minimum AT is 2.2 °C, appearing at 8:00; the AT rises fastest within 9:00–12:00, and it drops fastest within 16:00–21:00, which is related to the change in solar radiation intensity.

Figure 22 displays the measured values of wind speed (*v*) and roof temperature under brick (RTWB) in the winter. As can be observed from the figure, it is basically wind-free from 0:00 to 10:00, and the RTWB continues to decline from 10.3 °C to 8.9 °C. Within 11:00–13:00, the RTWB is stable at 8.9 °C, and later begins to continue to rise, reaching the highest temperature at 23:00. When the wind speed is approximately 0 m/s, the RTWB drops continuously, but the RTWB rises when the wind speed increases, which indicates that the increase in wind speed does not play an obvious cooling effect. The temperature transfer of the water-retained brick roof is delayed; therefore, the influence of wind speed on the RTWB has no obvious regularity.

Figure 23 shows the measured values of roof temperature under brick (RTWB) and solar radiation intensity (*I*) in the winter. As can be seen from the figure, the RTWB still decreases during the period when the solar radiation continues to rise. The solar radiation intensity is the highest at 14:00 and 15:00, which are 638 W/m^2^ and 663 W/m^2^, respectively, and thus the RTWB at both moments rises.

Figure 24 illustrates the measured characteristic temperatures in the winter. According to the figure, the brick bottom temperature (BT) within 24 h is significantly lower than the roof temperature under the brick (RTWB), indicating that the roof transfers heat up, and the water-retained brick plays an insulation effect. The water temperature (WT) is always lower than the bottom temperature of the brick (BT) from 0:00 to 12:00, indicating that the heat continuously transfers from the bottom of the brick during this period, and the WT increases due to the solar radiation. The WT starts to conduct heat downward from 13:00 when it is higher than BT. At this time, the temperature difference is very small, and the maximum temperature difference is 1.1 °C. Compared with the air, the specific heat capacity of water is large, so the water temperature drops from 17:00; and due to inertia and delay, BT drops from 19:00, delayed by two hours. From 19:00, BT> WT, water heat storage works against the night sky cold radiation; WT <BT <RTWB, this means that the heat dissipation of the penthouse is still decayed, thus the RTWB locates at a high temperature level.

Figure 25 shows the measured values of AT, RTWB and RT in the winter. It can be seen from the figure that the change in RT and AT with time is close with each other; however, the maximum value of them differs a lot when the time is 14:00. Furthermore, the average RT is 10.9 °C throughout the day, and the temperature fluctuation range is 2.5–23.8 °C. The maximum temperature fluctuation is 21.3 °C, and the minimum temperature occurs at 8:00. RTWB has an average of 9.6 °C throughout the day, with a temperature fluctuation range of 8.9–10.3 °C, and a maximum temperature fluctuation of 1.4 °C. The periods when the RT is lower than RTWB are 0:00–0:00 and 21:00–23:00, accounting for 54% of the whole duration, and all of them are in the night when the temperature is low. This indicates that the thermal insulation effect of the ordinary roof is poor, and indoor temperature fluctuation is bound to be large, hence, the thermal comfort of the ordinary roof room is poor, especially at night.

Figure 26 reveals the measured values of CTWB and CT in the winter. It can be seen from the figure that the change in CT and CTWB with time is similar with each other; however, the minimum value of them differs a lot when the time is 8:00. Moreover, the average CTWB is 11.0 °C, the maximum CTWB is 12.8 °C, the minimum CTWB is 9.5 °C and the maximum fluctuation of CTWB is 3.3 °C for 24 h on the sunny day in the winter. The average CT is 8.7 °C, the maximum CT is 12.9 °C, the minimum CT is 5.4 °C, and the maximum fluctuation of CT is 7.5 °C. Although the maximum ceiling temperature of the rooftop with bricks after laying water-retained bricks is the same and even lower than that without laying bricks, the average ceiling temperature of the rooftop with bricks in 24 h can be increased by 2.3 °C, and the fluctuation of the ceiling temperature of the rooftop with bricks is decreased by 56.0%. It can be seen that laying water-retained bricks improves the insulation performance of the roof and indoor thermal comfort.

To sum up, after laying water-retained bricks, the average ceiling temperature of the rooftop with bricks can increase by 2.3 °C in 24 h in the sunny day in the winter, and the fluctuation in the ceiling temperature of the rooftop with bricks can decrease by 56.0%. Both the thermal insulation performance and the indoor thermal comfort can be improved.

## 5. Conclusions

In this paper, the thermal insulation performance of the roof with water-retained bricks was first analyzed theoretically from the perspective of thermal inertia, attenuation and delay time. Then, the experimental rig was established to carry out the experimental investigation on the thermal insulation performance of the roof with and without water-retained bricks on the sunny, overcast and rainy days in the summer and on the sunny day in the winter. The main findings and future research development are as follows:(1)Due to the complex structure of the roof with water-retained bricks, whether the air interlayer is ventilated or not, the surface heat storage coefficient is affected by the evaporating heat transfer of the water layer. Therefore, the thermal insulation performance of the roof with water-retained bricks needs to be analyzed layer by layer.(2)When the water depth is 2 cm, the thermal inertness, attenuation and delay time of the roof with water-retained bricks are 2.575, 21 and 6.94 h, respectively. Theoretical calculation indicated that laying water-retained bricks can improve the heat insulation performance of the rooftop and the thermal comfort of the penthouse.(3)The experimental results indicated that after laying water-retained bricks, on the sunny day in the summer, the maximum temperature in the penthouse can be reduced by 2.3 °C, the average temperature in 24 h can be reduced by 0.6 °C, and the temperature fluctuation in the penthouse can be reduced by 52.5%. On the overcast day in the summer, the temperature fluctuation in the penthouse decreases by 60.0%. On the rainy days in summer, the temperature fluctuation in the penthouse decreases by 31.1%. That is to say, on the sunny, overcast and rainy days, laying water-retained bricks can enhance the heat insulation performance of the roof, and can improve the thermal comfort of the penthouse.(4)The experimental results also showed that after laying water-retained bricks, although the maximum temperature of the penthouse in the sunny day of the winter is the same as that without laying bricks, and even slightly lower than that without laying bricks, the average temperature of the penthouse in 24 h increases by 2.3 °C, and the temperature fluctuation of the inside penthouse decreases by 56.0%. Therefore, the heat insulation effect is significantly improved after laying water-retained bricks.

The future research development: the quantitative analysis on the influence of wind speed on the water evaporation and associated thermal insulation of the roof with water-retained bricks will be conducted in the future studies.

## Figures and Tables

**Figure 1 entropy-24-01528-f001:**
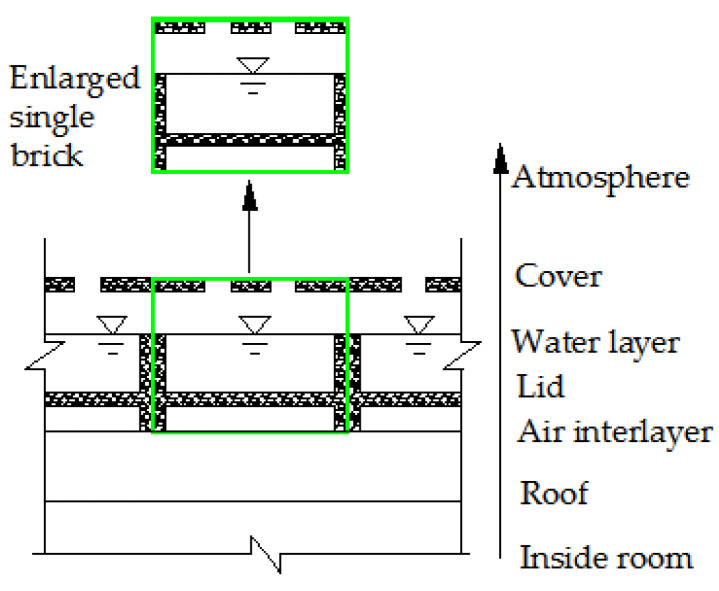
Cross-sectional schematic of the roof with water-retained bricks.

**Figure 2 entropy-24-01528-f002:**
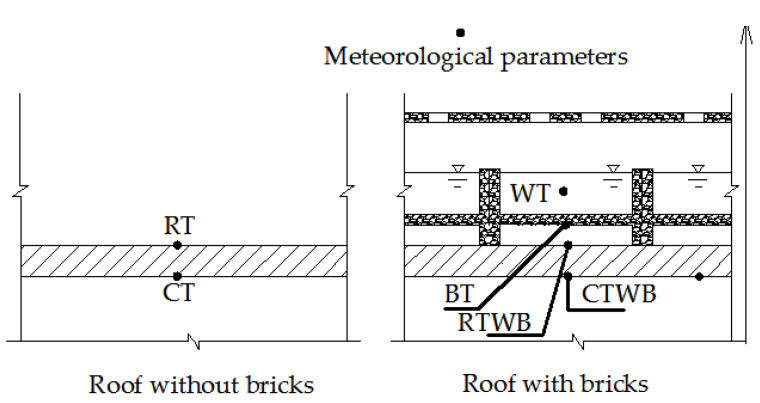
The schematic of the sensor distribution.

**Figure 3 entropy-24-01528-f003:**
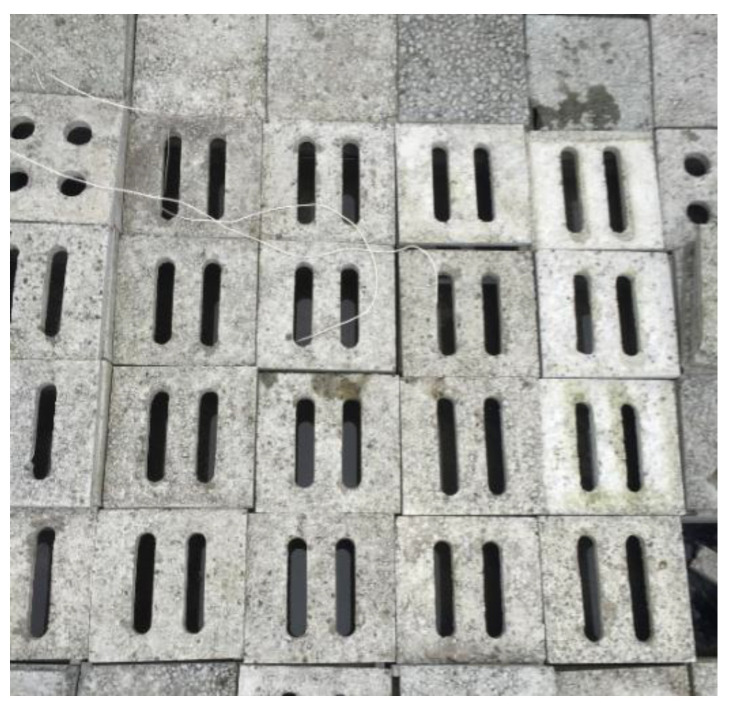
The photo of the field test.

**Figure 4 entropy-24-01528-f004:**
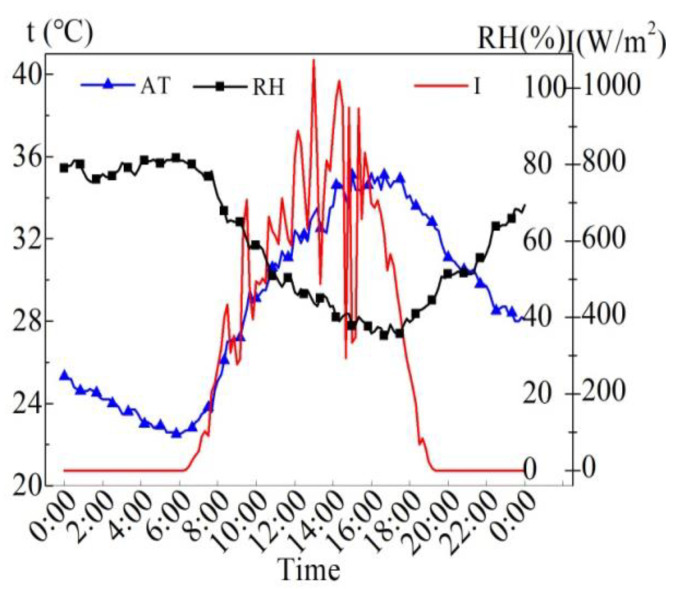
Meteorological parameters on August 6, 2015.

**Figure 5 entropy-24-01528-f005:**
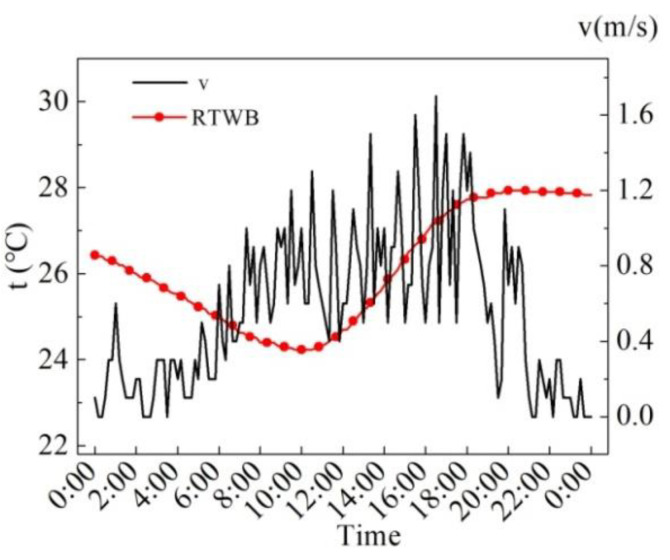
Measured *v* and RTWB on the sunny day in summer.

**Figure 6 entropy-24-01528-f006:**
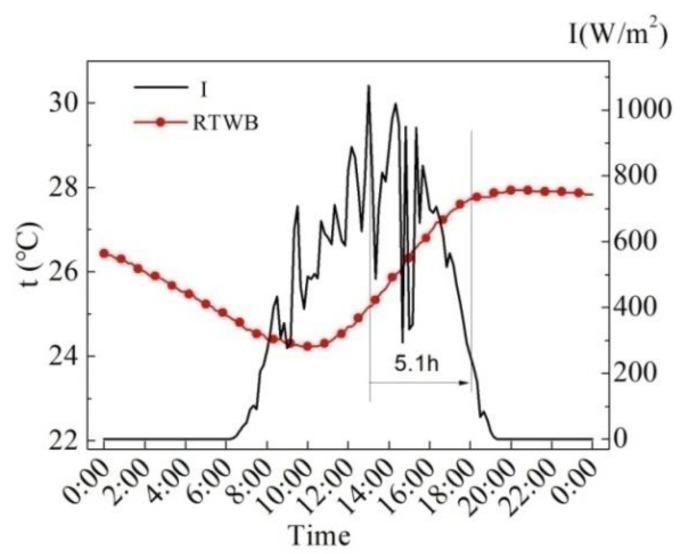
Measured *I* and RTWB on the sunny day in the summer.

**Figure 7 entropy-24-01528-f007:**
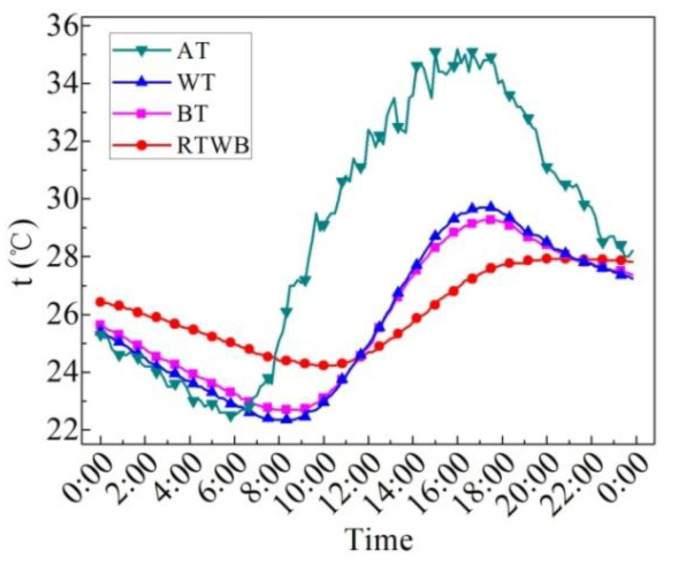
Measured characteristic temperatures of the roof with water-retained brick on the sunny day in the summer.

**Figure 8 entropy-24-01528-f008:**
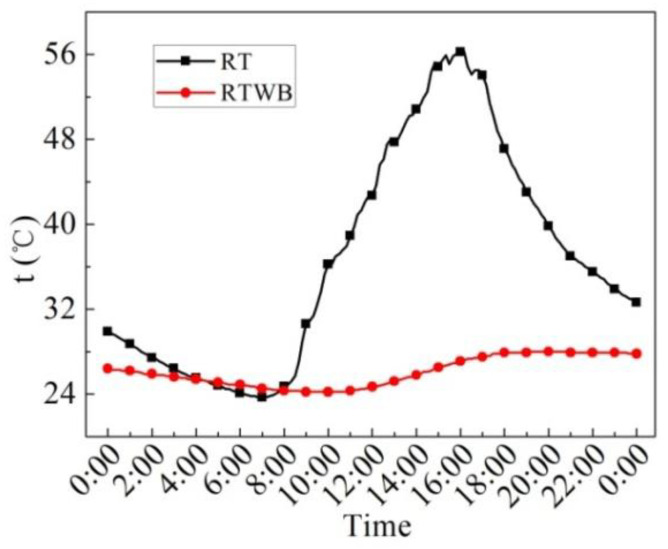
Measured values of RTWB and RT on the sunny day in the summer.

**Figure 9 entropy-24-01528-f009:**
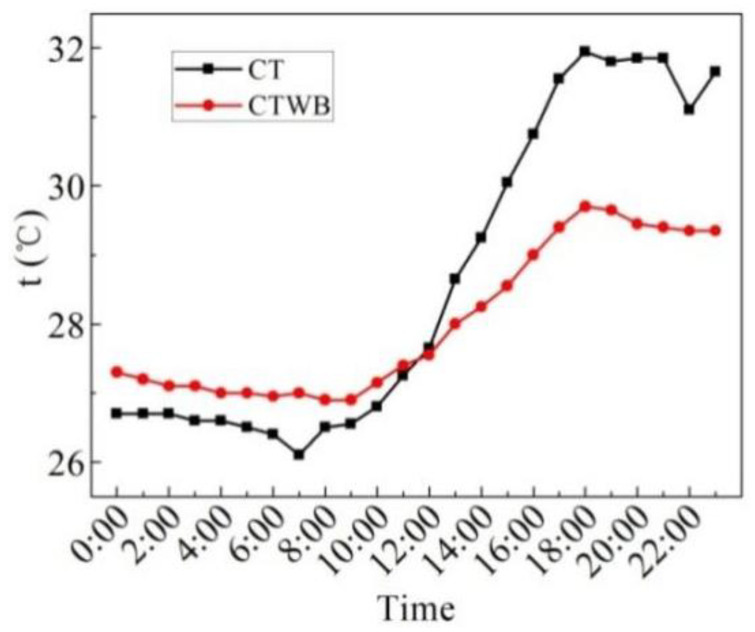
CTWB and CT measured on the sunny day in the summer.

**Figure 10 entropy-24-01528-f010:**
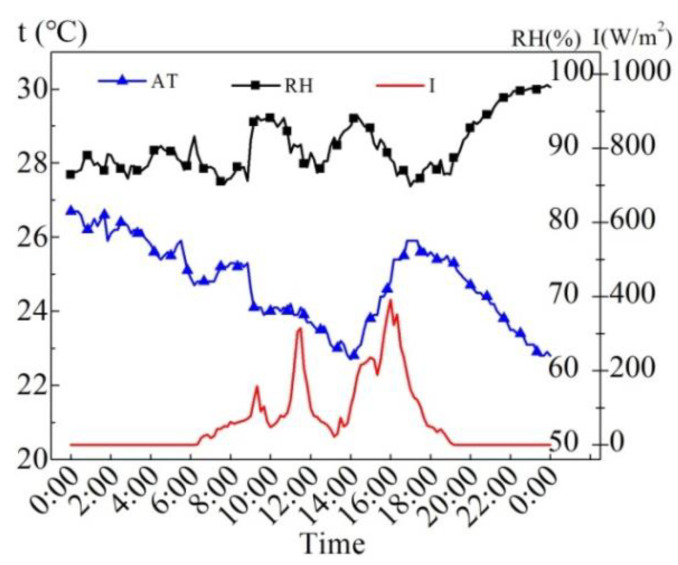
Measured meteorological parameters on the overcast day in the summer.

**Figure 11 entropy-24-01528-f011:**
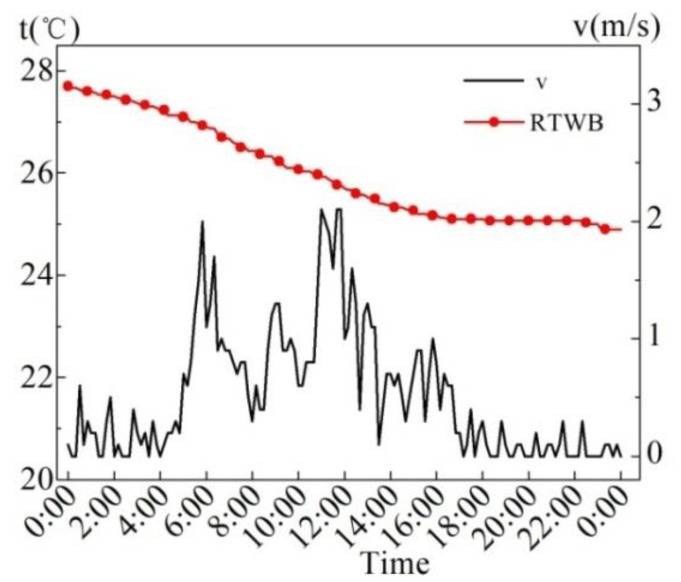
Measured values of *v* and RTWB on the overcast day in the summer.

**Figure 12 entropy-24-01528-f012:**
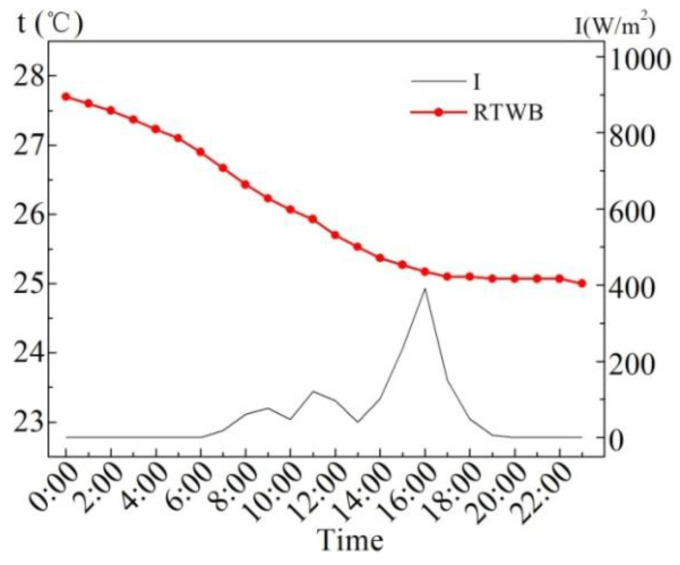
Measured values of *I* and RTWB on the overcast day in the summer.

**Figure 13 entropy-24-01528-f013:**
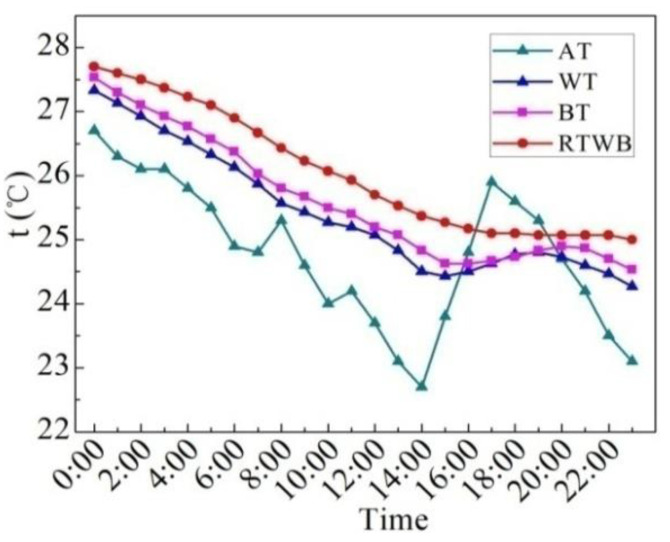
Measured characteristic temperatures on the overcast day in the summer.

**Figure 14 entropy-24-01528-f014:**
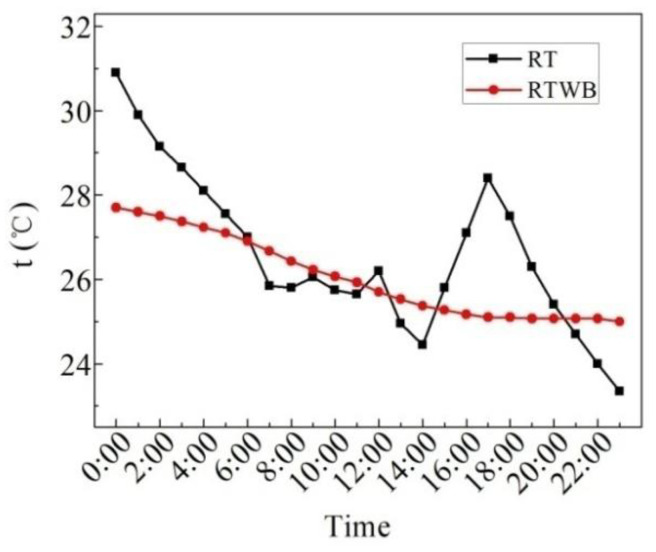
Measured RTWB and RT on the overcast day in the summer.

**Figure 15 entropy-24-01528-f015:**
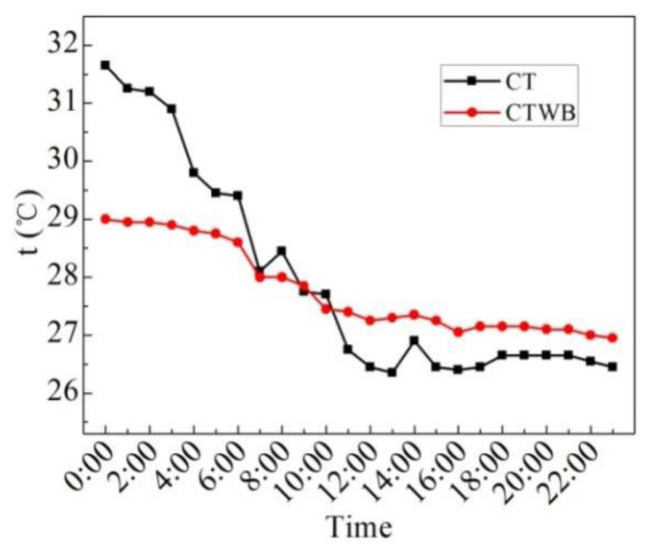
CTWB and CT measured on the overcast day in the summer.

**Figure 16 entropy-24-01528-f016:**
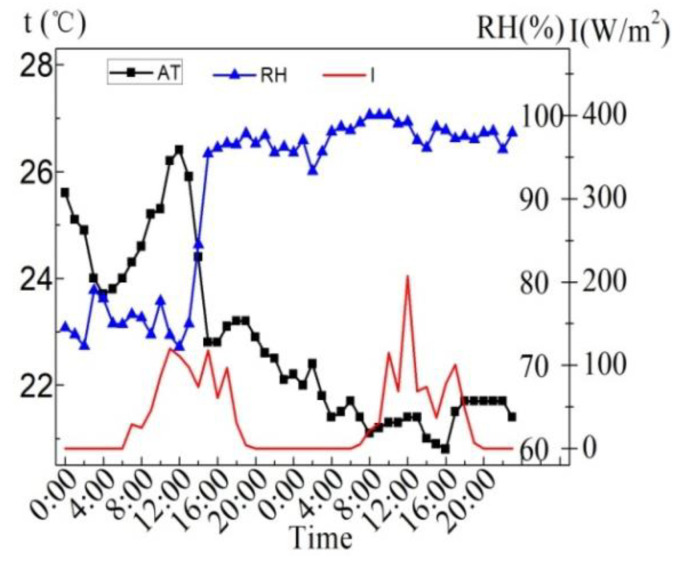
Meteorological parameters on the rainy days in the summer.

**Figure 17 entropy-24-01528-f017:**
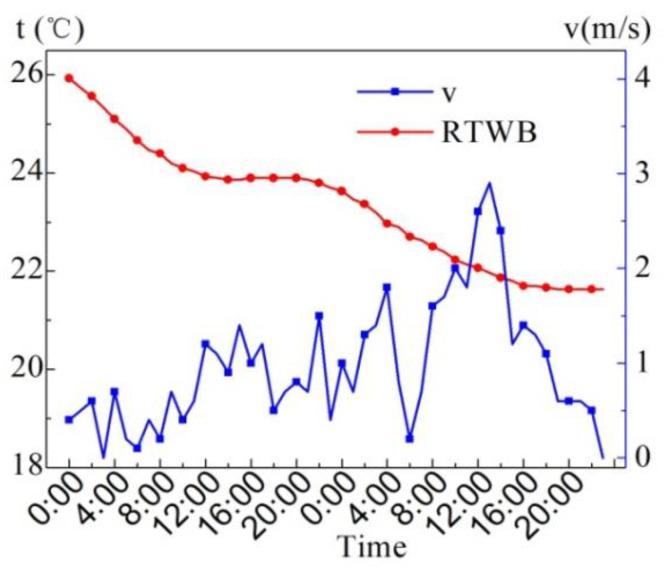
Measured *v* and RTWB on the rainy day in summer.

**Figure 18 entropy-24-01528-f018:**
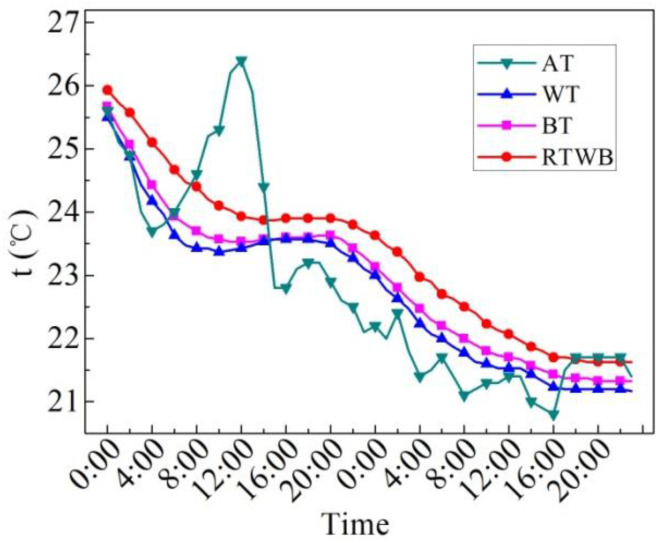
Characteristic temperatures measured on the rainy day in the summer.

**Figure 19 entropy-24-01528-f019:**
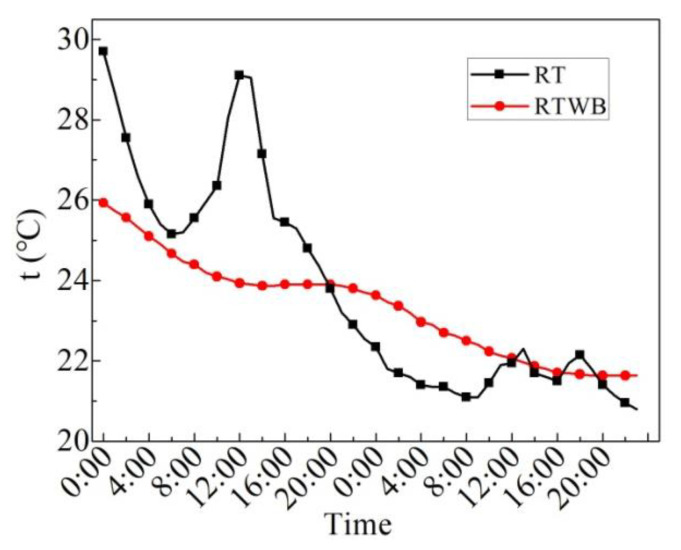
RTWB and RT measured on the rainy day in the summer.

**Figure 20 entropy-24-01528-f020:**
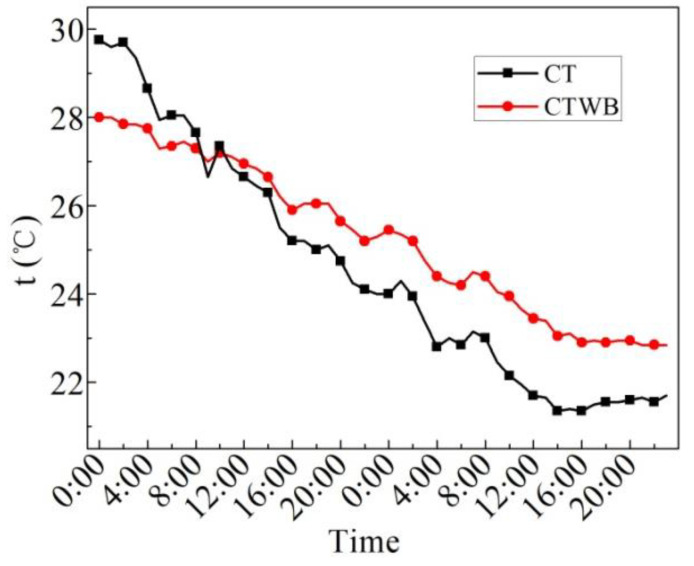
CTWB and CT measured on the rainy day in the summer.

**Figure 21 entropy-24-01528-f021:**
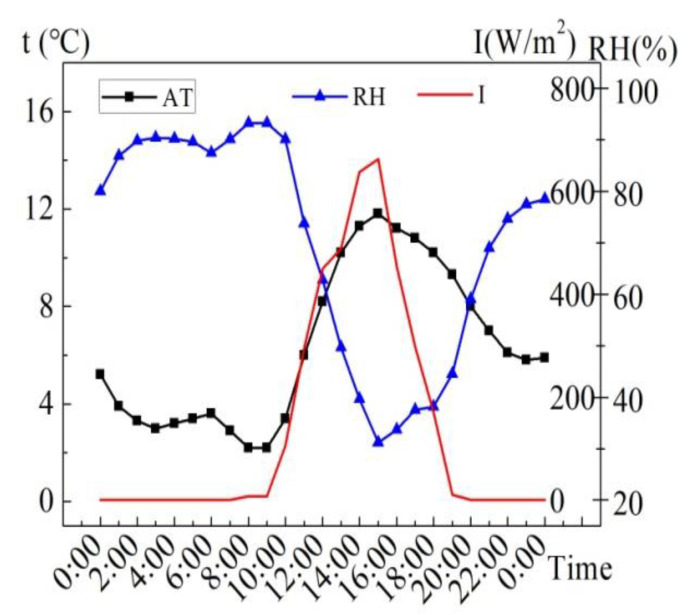
Meteorological parameters on the sunny day in the winter.

**Figure 22 entropy-24-01528-f022:**
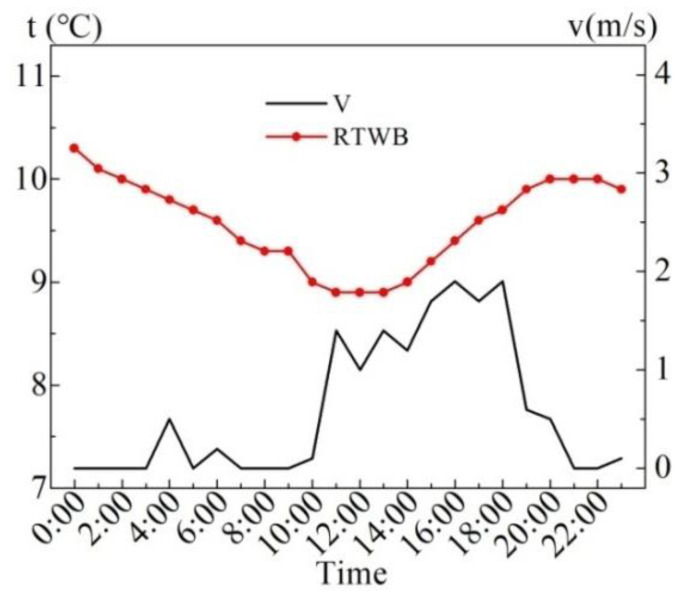
RTWB and *v* measured on the sunny day in the winter.

**Figure 23 entropy-24-01528-f023:**
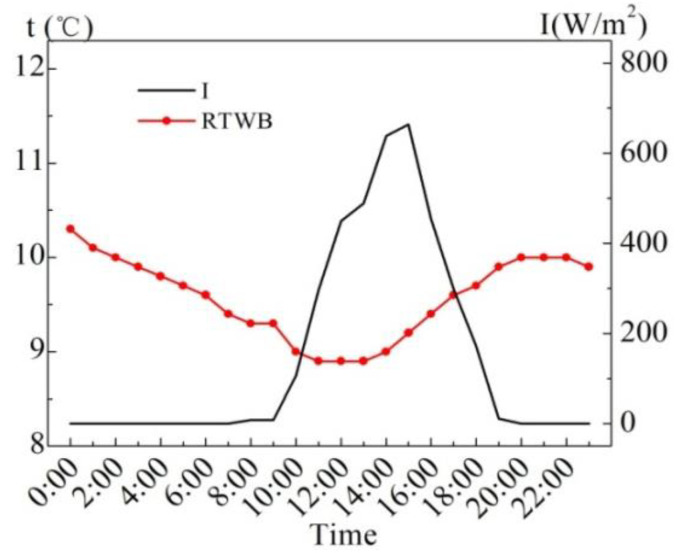
RTWB and *I* measured on the sunny day in the winter.

**Figure 24 entropy-24-01528-f024:**
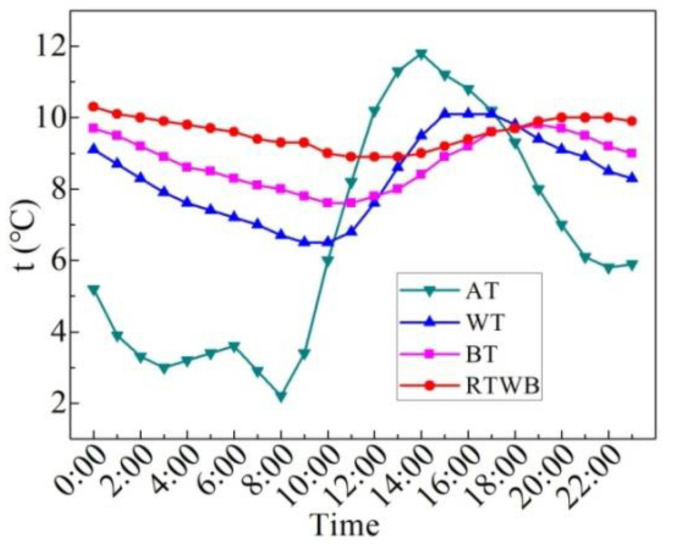
Characteristic temperatures measured on the sunny day in the winter.

**Figure 25 entropy-24-01528-f025:**
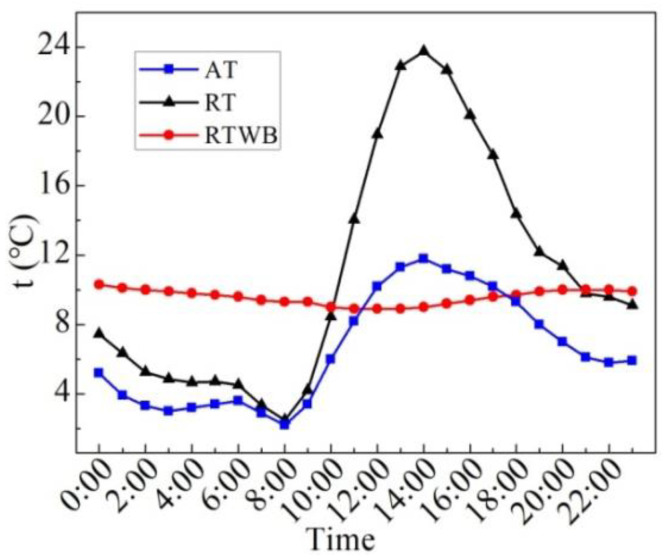
AT, RTWB and RT measured on the sunny day in the winter.

**Figure 26 entropy-24-01528-f026:**
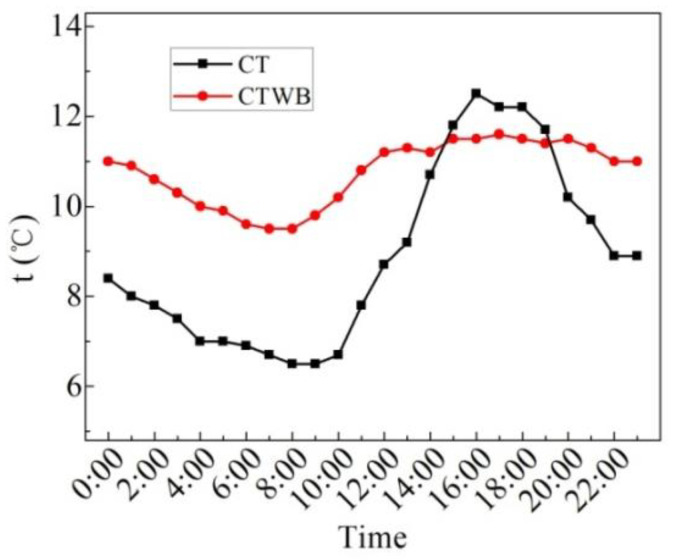
CTWB and CT measured on the sunny day in the winter.

**Table 1 entropy-24-01528-t001:** Thermal and physical parameters of each layer of the roof with water-retained bricks.

Serial Number	Material Layer	d (m)	λ (W/(m·°C))	R ((m^2^·°C)/W)	$S_{24}\;(W/(m^{2}\cdot {}^{\circ}C))$	D	Remarks
①	Lime mortar	0.02	0.81	0.025	10.07	0.249	
②	Reinforced concrete	0.1	1.74	0.057	17.2	0.980	
③	Cement mortar	0.02	0.93	0.022	11.37	0.245	
④	Air interlayer	0.02	0.0263	0.14 (Recommended)	0.048	0.007	Ventilation is considered
⑤	Lid	0.02	0.504	0.04	7.938	0.327	
⑥	Water	0.12 0.10 0.08 0.06 0.04 0.02 0	0.6	0.2 0.1667 0.133 0.1 0.067 0.033 0	13.5	2.7 2.25 1.8 1.35 0.9 0.45 0	1.0 is the critical value
⑦	Air layer	Water vapor 0.012	0.25	0.048	0.036	0.002	
		Air 0.048	0.0263	0.14	0	0	
⑧	Cover	0.02	0.412	0.053	5.953	0.315	

**Table 2 entropy-24-01528-t002:** Thermal inertia, attenuation and delay time of roof with water-retained brick under different water depths.

H cm	R (m^2^ °C)/W	$D_{w}$	∑D	*ν* _0_	$\xi_{0}$ *h*
0	—	—	2.131	15	5.74
2	0.033	0.45	2.575	21	6.94
4	0.067	0.9	3.025	31	8.16
6	0.1	1.35	3.475	43	9.37
8	0.133	1.8	3.925	59	10.59
10	0.167	2.25	4.375	81	11.80
12	0.2	2.7	4.825	112	13.00

**Table 3 entropy-24-01528-t003:** Sensors with technical parameters.

Sensors	Resolution	Range	Accuracy
Solar radiation intensity	1 W/m^2^	0–2000 W/m^2^	≤5%
Temperature	0.1 °C	−50–80 °C	±0.3 °C
R.H.	0.1%	0–100%	±3%
Wind speed	0.1 m/s	0–70 m/s	±0.3 m/s

## Data Availability

Data available on request.

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
