# Peer review of "Theoretical and Experimental Study on the Thermal Insulation Performance of the Roof with Water-Retained Bricks"

_entropy, 2022, doi:10.3390/e24111528_

Round 1

Reviewer 1 Report

With considering the thermal inertia, attenuation and delay time of the roof with water-retained bricks, the authors analyzed the thermal insulation performance of the roof with water-retained bricks. In addition, the authors carried out experimental study on the thermal insulation performance of the roof with and without water-retained bricks on the sunny, overcast and rainy days in the summer and on the sunny day in the winter. The works of this paper is worthy to do, thus, it can be considered for publication if the authors address the following comments:

1. Please check the literature review of the “Introduction”, and remove references that are not so relevant to this study;

2. Please give more detailed descriptions of the water-retained brick;

3. Please more specific analysis on the results of the Section 2 with respect to the values of thermal inertia, attenuation and delay time obtained;

4. Further analysis for some results obtained in Section 4 should be presented;

5. The format or some typing details should be improved;

6. The language of English should be improved.

Reviewer 2 Report

In this paper the thermal insulation performance of the roof with water-retained bricks was first analyzed. The topic is worth-of-interest. However, I require some comments to be addressed before considering again this paper.

My comments are:

1.      The authors stated that “the surface heat storage coefficient is affected by the evaporating heat transfer of the water layer” among the results of their research. However, I did not find suitable information into the methodology and results of the paper. I ask the authors including more information on the role of evaporation and on the method used to evaluate it.

2.      I did not understand why the authors only focused on the thermal performance of 2-cm water layers if they varied the water thickness between 0 and 12 cm. The role of the water is central in this paper; therefore, I suggest the authors providing more information thermal performance due to the different water thickness into the brick.

3.      The authors repeated the text and the figures for each of the weather condition analyzed, thus making the reading of this paper quite long. I suggest the authors to summarize the discussion of the results and the figures. This can help the authors to better compare and analyze the results.

4.      Among the results, the authors mentioned that the solution applied can improve the thermal comfort. However, in my knowledge, they only evaluated the internal surface temperature of the loft that is one of the factors influencing the thermal comfort but nor the only one nor the most important. I require the authors to provide more information or to revise the results in terms of comfort.

5.      Finally, some limitations and future research developments are expected in the conclusion section, along with the importance of conducting this research for manufacturer and researcher. The authors just conclude the paper with the summary of the results.

Reviewer 3 Report

The authors investigated the thermal insulation performance of the roof with water-retained brickswith considering the thermal inertia, attenuation and delay time of the roof with water-retained bricks. Moreover,they performedexperimental study on the thermal insulation performance of the roof with water-retained bricks under different weather conditions.In my opinion, the research is full of scientific meanings.It is considered for the possible publication in the journal of “Entropy” if the authors can make revisions according to the following comments:

1. Typing errors should be avoided, e.g., the “4” in “R4” of line 182, it should be a subscript; please check other possible errors.

2. The language of English needs to be improved, e.g., the last second paragraph of the “Introduction”, it should be “However, no literature mentioned the theoretical and experimental study on the thermal insulation performance of the roof with water-retained bricks as illustrated in Figure 1.”.

3. Please give out more specific description of the experimental rig in the Section 3.

4. Please further elaborate the inter-relationshipbetween the Section 2 and Sections 3&4.

5. If possible, please give out deeper analysis for the Section 4.

Round 2

Reviewer 2 Report

The authors addressed all my comments. Thank you